# Predicting 1, 2 and 3 year emergent referable diabetic retinopathy and maculopathy using deep learning
Paul Nderitu [1,2] ✉, Joan M. Nunez do Rio[2,3], Laura Webster[4], Samantha Mann [4], M. Jorge Cardoso [3], Marc Modat[3], David Hopkins[5], Christos Bergeles [3,6] & Timothy L. Jackson[1,2,6]

## Abstract

**Background** Predicting diabetic retinopathy (DR) progression could enable individualised screening with prompt referral for high-risk individuals for sight-saving treatment, whilst reducing screening burden for low-risk individuals. We developed and validated deep learning systems (DLS) that predict 1, 2 and 3 year emergent referable DR and maculopathy using risk factor characteristics (tabular DLS), colour fundal photographs (image DLS) or both (multimodal DLS).

**Methods** From 162,339 development-set eyes from south-east London (UK) diabetic eye screening programme (DESP), 110,837 had eligible longitudinal data, with the remaining 51,502 used for pretraining. Internal and external (Birmingham DESP, UK) test datasets included 27,996, and 6928 eyes respectively.

**Results** Internal multimodal DLS emergent referable DR, maculopathy or either area-under-the receiver operating characteristic (AUROC) were 0.95 (95% CI: 0.92–0.98), 0.84 (0.82–0.86), 0.85 (0.83–0.87) for 1 year, 0.92 (0.87–0.96), 0.84 (0.82–0.87), 0.85 (0.82–0.87) for 2 years, and 0.85 (0.80–0.90), 0.79 (0.76–0.82), 0.79 (0.76–0.82) for 3 years. External multimodal DLS emergent referable DR, maculopathy or either AUROC were 0.93 (0.88–0.97), 0.85 (0.80–0.89), 0.85 (0.76–0.85) for 1 year, 0.93 (0.89–0.97), 0.79 (0.74–0.84), 0.80 (0.76–0.85) for 2 years, and 0.91 (0.84–0.98), 0.79 (0.74–0.83), 0.79 (0.74–0.84) for 3 years.

**Conclusions** Multimodal and image DLS performance is significantly better than tabular DLS at all intervals. DLS accurately predict 1, 2 and 3 year emergent referable DR and referable maculopathy using colour fundal photographs, with additional risk factor characteristics conferring improvements in prognostic performance. Proposed DLS are a step towards individualised risk-based screening, whereby AI-assistance allows high-risk individuals to be closely monitored while reducing screening burden for low-risk individuals.

## Plain english summary

Diabetic retinopathy (DR) is a disease where the light-sensing layer at the back of the eye (retina) becomes damaged by raised blood sugar levels. It affects around one in three of the 463 million people with diabetes worldwide and is a leading cause of acquired vision loss in working-age adults. In this study, we developed computer-based models to predict when DR would reach a stage where vision could be threatened up to 3-years in the future. Our study shows that this system can accurately predict sight-threatening DR in patients with diabetes. This could mean fewer unnecessary visits for individuals at low-risk of DR progression, but closer monitoring and potentially earlier treatment for individuals at high-risk of DR progression, which could reduce the risk of vision loss.

The prevalence of diabetes mellitus (DM) is estimated to rise rapidly from 463 million people in 2019, to 700 million by 2045[1]. This 51% increase will result in a marked rise in the prevalence of diabetic retinopathy (DR) which affects about one in three people with DM and is a leading cause of vision loss in working age adults[2]. Globally, DR is the fifth most common cause of

blindness and the only one with an increasing age-standardised prevalence (+14.9%) between 1990 and 2020 in adults aged 50 years and older[3].

The purpose of the UK's Diabetic Eye Screening Programme (DESP) is to detect referable DR using colour fundal photographs (CFPs). The UK DESP screens around 3.2 million people with diabetes every year, and is

[1]Section of Ophthalmology, Faculty of Life Sciences and Medicine, King's College London, London, UK. [2]Department of Ophthalmology, King's Ophthalmology Research Unit (KORU), King's College Hospital, London, UK. [3]School of Biomedical Engineering & Imaging Sciences, King's College London, London, UK. [4]Department of Ophthalmology, South East London Diabetic Eye Screening Service, St Thomas' Hospital, London, UK. [5]Institute of Diabetes, Endocrinology and Obesity, King's Health Partners, London, UK. [6]These authors jointly supervised this work: Christos Bergeles, Timothy L. Jackson. ✉e-mail: p.nderitu@doctors.org.uk

staffed by over 1500 highly trained human graders[4]. Currently, trained graders assess for DR severity using CFPs, and they examine over 13 million images per year[4]. The majority (~95%) of individuals attending screening have non-referable DR and are reviewed annually[5]. A small proportion of individuals (~5%) are referred to the hospital eye service for closer monitoring or treatment[5]. Screening is very cost, time and labour intensive but allows for the detection of sight-threatening DR amenable to early treatment[4]. Screening is credited for the significant reduction in DR-related sight-impairment registrations in the UK[6]. DR screening has also been directly credited for a reduction in the prevalence of blindness in other countries including Sweden, Iceland, and Poland[7].

The key decision point within DR screening that determines if further intervention is required is the emergence of referable disease. Identifying individuals at-risk of progressing to referable disease could allow for the prompt initiation of preventative treatments that reduce the risk of sight-loss. Concurrently, screening intervals could be extended for low risk individuals, with 80% of attendees being at a low-risk of progression to referable DR[8,9]. Therefore, a prognostic model that predicts future emergent referable disease could allow for the individualisation of screening intervals and preventative interventions that are commensurate to the level of risk. A randomised controlled trial reported that individualised risk-based screening could reduce appointment requirements by over 40%, and service costs by 20%[8,9]. Notably, attendee satisfaction was equivalent to annual screening, with no statistically significant differences in attendance or detection of emergent referable DR[8,10,11].

A systematic review described 14 non-AI based DR progression risk prediction models, which were all based on systemic risk factors or historical retinopathy grades[12–18]. Systemic risk factor models frequently require HbA1c and lipid biochemistry, necessitating invasive blood tests which are not be contemporaneously collected by DR screening services[12,13,17,18], requiring complex data linkage or imputation to implement[12]. These prerequisites substantially complicate the deployment and deliverability of such models[8,12]. Retinopathy-grade based approaches have low specificity for detecting low-risk individuals, which results in a higher rate of unnecessary hospital referrals, limiting their cost-effectiveness[14]. Additionally, retinopathy-grade approaches usually require individuals attend two sequential visits to make a prediction[15,16], consuming resources and being reliant on attendance, yet 10–15% of DESP appointments are missed[19]. A recent study by Olvera-Barrios et al. also found that retinopathy-grade based risk-stratification approaches could delay the detection of sight-threatening DR, particularly for patients of black ethnic origin[20].

A feasibility study established that deep learning systems (DLS) can predict DR progression using CFPs, but this study included only non-representative clinical trial participants and had a very small sample size (499 patients)[21]. Bora et al. developed a DLS to predict any DR onset within 2 years using CFPs, however, their model explicitly excluded patients with mild DR (~15%) who still attend screening annually hence the model would not be applicable to them[22]. Recent studies have used DLS to predict progression to referable DR using CFPs but have not explored the prediction of emergent referable maculopathy independently which is important as the risk to sight-loss differs significantly to that of emergent referable DR (e.g., proliferative DR)[23,24].

In this study, we train DLS to simultaneously predict future emergent referable DR, referable maculopathy, or either outcome (referable DR or maculopathy) using non-invasive, single-visit screening data. Independent DLS are developed and validated for each of the 1, 2 and 3 year prediction intervals using non-invasive risk factor characteristics alone (tabular DLS), two-field CFPs alone (image DLS) or both (multimodal DLS). Internal and external validation is performed using data from two geographically distinct UK DESPs. We assess DLS performance stratified by age, sex and ethnicity, and at the individual-level. Finally, ablation and attribution studies are performed to ascertain CFPs features and risk factor characteristics pertinent to DLS predictions.

## Methods

### Study population and datasets

This study was conducted in accordance with the tenets of the Declaration of Helsinki and TRIPOD guidelines (Supplementary pp 41). Health Research Authority approval and a favourable opinion from the East Midlands Leicester South Research Ethics Committee (REC, 20/EM/0250) were obtained prior to study commencement. Licensed access for the external dataset (INSIGHT, Health Data Research UK) was approved by the West of Scotland REC 4 (20/ES/0087). The need for individual consent was waived by the favourable ethical opinions for the use of pseudoanonymised retrospective data. Individuals aged ≥12 years attending screening between September 2013 to December 2019 (6.25 years) were eligible for inclusion. DESP procedures including CFP capture, grading and data collection were performed by trained graders using UK screening protocols and DR and maculopathy classification definitions[25] (R and M grades) (Supplementary Tables S1 and S2).

Extracted data were DR grades, maculopathy grades, CFPs and risk factor characteristics (age, physician-reported sex, self-reported ethnicity, DM type, DM duration, visual acuity, index of multiple deprivation rank)[26]. Development and internal test datasets were derived from the south-east London DESP (SEL-DESP, 27 sites), and external test datasets from the Birmingham, Solihull and Black Country DESP (BSBC-DESP, 110 sites; Fig. 1). Ineligible CFPs were removed with the assistance of curation DLS which we developed and validated previously and describe elsewhere (Supplementary methods pp 3)[27]. After removing ineligible CFPs, 1,181,858 gradable two-field CFPs from 102,446 SEL-DESP individuals (96.1%) and 66,286 gradable two-field CFPs from 4772 BSBC-DESP individuals (96.1%) were selected (Fig. 1). The SEL-DESP dataset was randomly divided, 80% for development (88% training, 12% tuning) and 20% for internal testing, with data from an individual in one partition exclusively, whilst 100% of BSBC-DESP data were used for external testing. Data curation, DESP procedures and CFP specifications are described in the Supplementary Methods pp 3 and Ssupplementary Table S2.

**Selection of pretraining and longitudinal datasets**. A valid longitudinal pair of visits, termed the baseline and predict visit, were those spaced 1, 2 or 3 years apart within a ± 2, 4 or 6 month tolerance respectively (Fig. 2, Supplementary Figs. S3A and S3B). Starting from the most recent attendance, the sequence of all visits per eye were searched to find the first valid baseline and predict pair, and eyes were only excluded from longitudinal modelling if all possible pairings were nonvalid. Baseline visits with referable DR or maculopathy were ineligible as emergent referable disease is the future prediction outcome. Eyes with a valid pair of visits were used to create 1, 2 or 3 year interval longitudinal dataset cohorts. Development set eyes ineligible for inclusion in all the 1, 2 or 3 year longitudinal cohorts were used for DLS pretraining on a cross sectional disease severity classification task (Fig. 1). The distribution of visits and inter-visit intervals are provided in the Supplementary Methods pp 3.

### Outcomes

DLS were trained to simultaneously predict three binary outcomes, (1) emergent referable DR (R2[+]) which included emergent moderate non-proliferative or worse DR, (2) emergent referable maculopathy (M1) which included exudates or microaneurysms near the fovea or a group of exudates in the macula, or (3) either emergent disease (R2[+] | M1) (Fig. 2, Supplementary Figs. S3A and S3B, Supplementary Table S1). The outcome was considered present if it occurred at any point between the baseline and predict visit for the respective interval. Separate DLS were developed and evaluated for each of the 1, 2 and 3 year prediction intervals. Individuals with sufficient follow-up could contribute to all intervals independently, but only to either the train, tune, internal or external test partition per prediction interval.

## Model development

The image DLS comprised of two EfficientNet-V2-s models[28] (one per field, no weight sharing) with concatenation of the pre-classification feature maps. The tabular DLS was based on TabNet[29], a high-performance tabular data neural network. Justification for the chosen image and tabular DLS

models are provided in the Supplementary pp 4. The tabular DLS had random weight initialisation (as no relevant pretrained model) while the image DLS was initialised using ImageNet weights. Tabular and image DLS were first pretrained to classify DR severity (R0: no DR, R1: mild-moderate DR, R2: moderate-severe DR, R3a: proliferative DR) and maculopathy

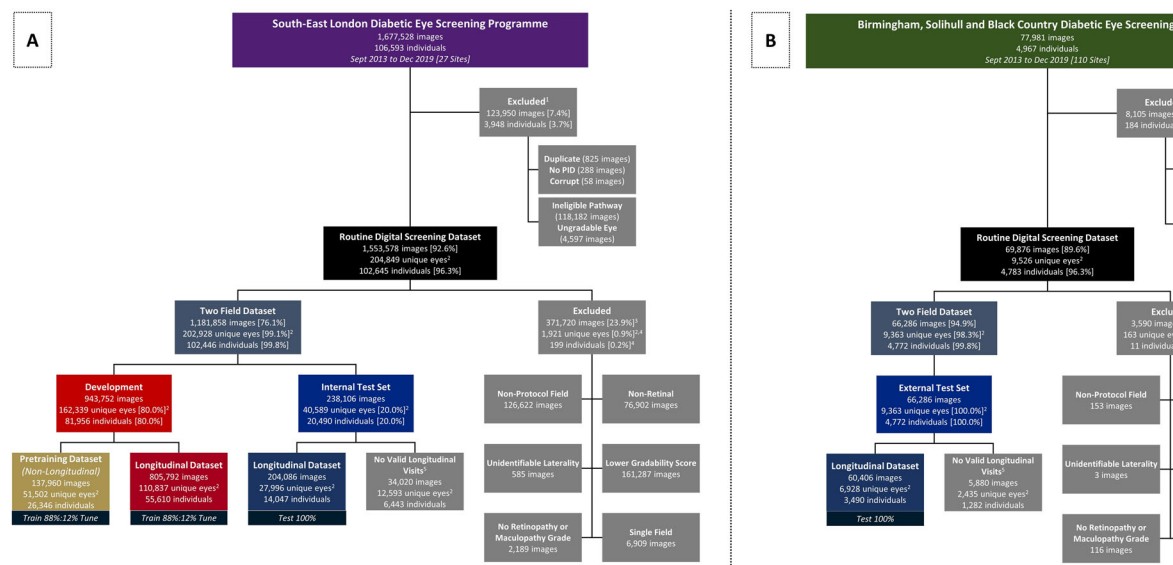

**Fig. 1 | Study datasets.** Study flow chart illustrating the formation of the development and internal test sets from the southeast London diabetic eye screening programme (**A**) used to for the pretraining and longitudinal datasets, and the external test set from the Birmingham, Solihull and Black Country diabetic eye screening programme (**B**). Cohorts for the 1, 2 and 3 year prediction intervals are created from the 'Longitudinal Datasets' and described in detail in Supplementary Data. 1.

Ineligible pathway visits were digital surveillance (attendance or referral), non-DR hospital eye service referral, or slit-lamp biomicroscopy (attendance or referral). [1]Sequential exclusion from duplicates to ungradable eyes. [2]Eyes are counted only once. [3]Overlapping groups hence group sum may not equal the image total. [4]Counted if no images are present in the two field dataset. [5]Samples not used and are therefore excluded. PID=Patient identifier.

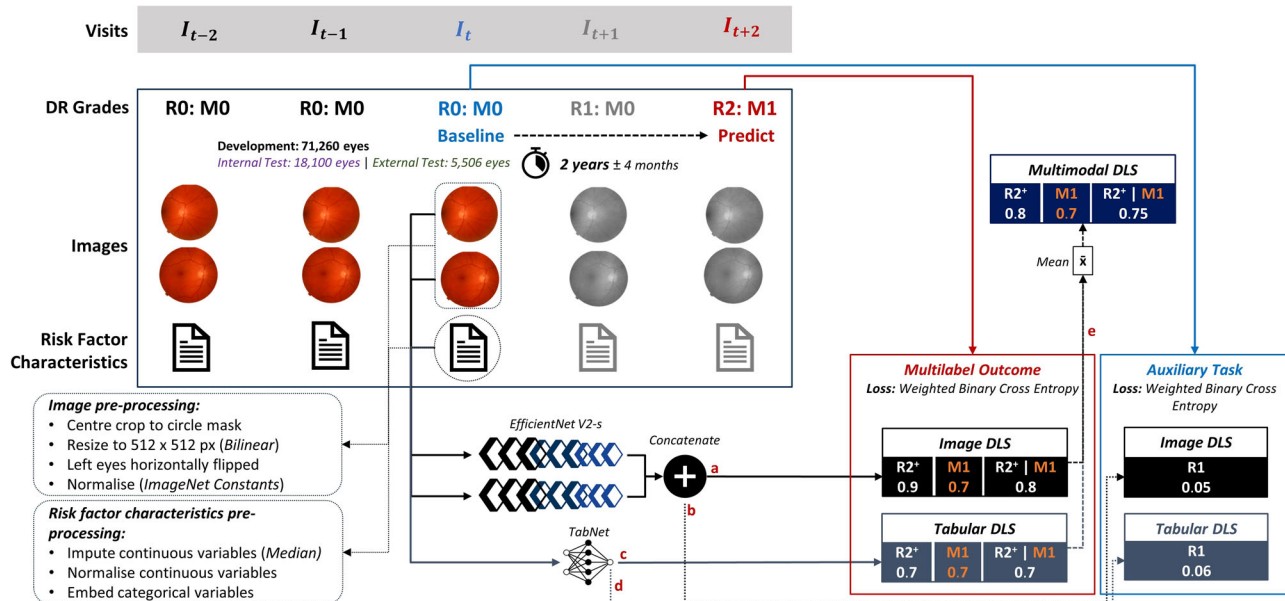

**Fig. 2 | Deep learning system training schema for the 2 year prediction interval.** Data pre-processing and training schema used to develop DLS for the 2-year prediction interval. Two-field images from the baseline visit (blue) are used as input to the image DLS, whilst characteristics are inputs for the tabular DLS. Baseline and predict visits (red) must be 2 years from the baseline visit ±4 months. DLS predict if emergent referable disease occurs between the baseline and predict visit. Image and tabular DLS are trained to predict the binary progression outcomes (a & c) with the ground truth from the predict visit (red continuous line), with an auxiliary task of

detecting baseline DR (b & d) with the ground truth from the baseline visit (blue continuous line). Numbers below each outcome indicative of the prediction score. The multimodal DLS is formed by taking a mean of the tabular and image DLS predictions at test time I. DLS=Deep learning system. $I_t$=Visit number in the sequence. R0=No DR. R1=Mild-moderate DR. R2=Moderate-severe DR. M0=No referable maculopathy. M1=Referable maculopathy. R2$^+$ | M1=Referable DR or maculopathy. R2$^+$=Referable DR.

(M0: no referable maculopathy, M1: referable maculopathy) as per UK DESP grading criteria (Supplementary Table S1, Supplementary Methods pp 4–5). After pretraining, tabular and image DLS were then trained to predict emergent referable DR, referable maculopathy or either, as a multilabel binary objective, with an auxiliary binary objective of detecting mild-moderate DR at the baseline visit. The auxiliary objective of detecting baseline mild-moderate DR was chosen because developing early DR is a significant risk factor for progression to referable DR[30]. Additionally, cross sectional detection of DR severity has been reported to improve DLS performance in predicting DR progression[22]. Both the primary and auxiliary objectives contributed equally to the balanced two-term loss function (Supplementary Methods pp 5, Fig. 2, Supplementary Figs. S3A and S3B). The multimodal DLS was created from the ensemble of the trained image and tabular DLS by computing the mean of their predictions at test time as early experiments showed this strategy was superior to end-to-end training of a single multimodal DLS by concatenating the final feature vectors of the image and tabular DLS (Supplementary Methods pp 6). Additional details on data pre-processing, missing data handling, DLS models, training augmentations, hyperparameters, pretraining and longitudinal training are provided in the Supplementary Methods pp 3–6 and Supplementary Tables S4 and S5.

## Statistical analysis

Analyses were performed between November 2022 and June 2023 using SPSS v27 (IBM) and Python v3.9. Receiver operating characteristic curves (ROC) and AUROC were used to summarise DLS discriminant performance, with stratification by age, sex, and ethnicity. AUROC 95% confidence intervals and significance were calculated using DeLong's method. To estimate individual-level performance, the maximum of the predictions from both eyes was assessed against the 'worst' respective emergent referable disease outcome from both eyes. This eye-to-individual-level aggregation method was chosen because referable disease in either eye would result in a referral to the hospital eye service. The prediction and grade for individuals with only one eye were used for individual-level AUROC analyses. Bonferroni correction to $p < 0.017$ was applied given the three predefined emergent referable disease outcomes, namely referable DR ($R2^+$), referable maculopathy (M1) and either ($R2^+ |$ M1). Specificities at 80% sensitivity, a threshold chosen to be comparable to a prior UK systemic risk factor model which evaluated the performance of a systemic risk factor based prognostic model in individualised DR screening as part of a randomised control trial[12] were determined for comparability. We also computed positive and negative predictive value curves as a function of threshold, with 95% confidence intervals computed using the Clopper-Pearson method (β distribution). False negatives as a function of threshold, stratified by the predict visit DR or maculopathy grade were obtained to assess the risk of incorrect DLS predictions. Image ablation and attribution studies were performed to localise CFP regions and tabular characteristics associated with DLS predictions (Supplementary Methods pp 6). All CFPs shown in the figures are of a standard scale as captured by 45 degree imaging.

## Reporting summary

Further information on research design is available in the Nature Portfolio Reporting Summary linked to this article.

## Results

### Baseline characteristics

Using a development dataset of 110,837 eyes, tabular, image and multimodal DLS were trained to predict 1, 2 and 3 year emergent referable DR, referable maculopathy or either, with validation performed on 27,996 and 6928 eyes from the internal and external test datasets respectively (Fig. 1). Baseline characteristics, eyes per interval and referable disease rates per partition are shown in Supplementary Data 1. Baseline characteristics for eyes used in DLS pretraining or excluded are presented in supplementary table S4.

## DLS performance

Internal test multimodal DLS area-under-the receiver operating characteristic (AUROC) for predicting emergent referable DR, maculopathy or either were 0.95 (95% CI: 0.92–0.98), 0.84 (0.82–0.86), 0.85 (0.83–0.87) for 1 year; 0.92 (0.87–0.96), 0.84 (0.82–0.87), 0.85 (0.82–0.87) for 2 years, and 0.85 (0.80–0.90), 0.79 (0.76–0.82), 0.79 (0.76–0.82) for 3 years (Table 1 and Fig. 3). External test multimodal DLS AUROC for predicting emergent referable DR, maculopathy or either were 0.93 (0.88–0.97), 0.85 (0.80–0.89), 0.85 (0.76–0.85) for 1 year; 0.93 (0.89–0.97), 0.79 (0.74–0.84), 0.80 (0.76–0.85) for 2 years, and 0.91 (0.84–0.98), 0.79 (0.74–0.83), 0.79 (0.74–0.84) for 3 years.

Multimodal and image DLS AUROC were significantly better ($p < 0.017$) than tabular DLS in internal and external tests datasets (Table 1). There was a modest improvement in performance for the multimodal DLS compared to the image DLS (+0.01 to +0.06), driven largely by improved emergent referable maculopathy prediction. Prognostic AUROC was highest for 1-year intervals and lowest for 3-year intervals for all DLS, with minimal drop in performance (−0.02 to −0.06) for all outcomes on external testing (Table 1). DLS pretraining and the auxiliary objective of detecting mild-moderate DR at baseline during DLS training improved model validation performance (internal test AUROC + 0.2 to +0.5, supplementary methods pp 5). Individual-level results were similar to eye-level results for all outcomes and prediction intervals (supplementary table S6).

## Specificity, predictive value and subgroup analyses

At 80% sensitivity, multimodal DLS internal test specificity for detecting eyes at low-risk of emergent referable DR, maculopathy or either were 0.96, 0.69, 0.74 over 1-year, 0.92, 0.69, 0.70 over 2-years and 0.79, 0.58, 0.58 over 3-years respectively. External test multimodal DLS specificity at 80% sensitivity for detecting eyes at low-risk of emergent referable DR, maculopathy or either were 0.92, 0.76, 0.80 over 1-year, 0.93, 0.58, 0.60 over 2-years 0.87, 0.59, 0.59 over 3-years respectively (see Supplementary Table S7 for the full confusion matrices). From all progressing eyes, the percentage of eyes which were misclassified and progressed undetected to proliferative DR, moderate-severe non-proliferative DR or referable maculopathy in the internal vs external test datasets were 0.2% vs 0.0%, 1.0% vs 0.0% and 19.3% vs 20.1% for the 1-year interval, 0.3% vs 0.0%, 0.9% vs 0.0% and 19.0% vs 16.8% for the 2-year interval and 0.6% vs 0%, 1.5% vs 0.0% and 15.0% vs 13.6% for the 3-year interval. Most false negative cases were due to missed emergent referable maculopathy for all DLS (Supplementary Figs. S8A and S8B). Limited referable DR (<0.45%) and maculopathy (<2.3%) incidence rates (Supplementary Data 1) resulted in high negative predictive values, but constrained positive predictive values, both of which were better for the image and multimodal DLS (Supplementary Figs. S9A and S9B). Subgroup analysis whereby DLS performance was stratified by age, sex and ethnicity were performed, with results shown in Supplementary Data 2, Supplementary Data 3, and Supplementary Data 4. Where present, subgroup DLS underperformance (AUROC ≥ 0.10 vs mean) occurred in only one prediction interval per outcome in either the internal or external test set but not both.

## Ablation and attribution analyses

Ablation studies showed decreasing image DLS AUROC for combined emergent referable DR or maculopathy and also emergent referable maculopathy prediction with increasing masking of the macula field, whilst masking the nasal field had little effect on image DLS performance (Fig. 4). Macula field masking also reduced image DLS predictions of emergent referable DR but to a lesser extent, with little negative effect with nasal field masking. In support of the ablation findings, attributions maps show that image DLS attention is largely concentrated near the central macula for emergent referable DR and maculopathy prediction, with more variable areas highlighted in nasal fields (Supplementary Figs. S10, S11, S12, S13, S14, and S15). Attributions for sample eyes with incident disease are also provided in Supplementary Figs. S16, S17, and S18 for better localisation of pertinent image features.

**Table 1 | tabular, image and multimodal DLS performance**

| Model | Input Data | Outcome | Internal Test AUROC (95%CI) | | | External Test AUROC (95%CI) | | |
|---|---|---|---|---|---|---|---|---|
| | | | Year 1 Unique Eyes 25,154 | Year 2 Unique Eyes 18,100 | Year 3 Unique Eyes 14,327 | Year 1 Unique Eyes 6355 | Year 2 Unique Eyes 5506 | Year 3 Unique Eyes 4865 |
| Tabular DLS TabNet | Risk Factor Characteristics[a] | R2+ \| M1 | 0.72 (0.69–0.74) | 0.73 (0.71–0.76) | 0.70 (0.67–0.73) | 0.68 (0.63–0.72) | 0.69 (0.64–0.74) | 0.65 (0.60–0.70) |
| | | R2+ | 0.83 (0.78–0.87) | 0.76 (0.68–0.84) | 0.70 (0.63–0.77) | 0.73 (0.63–0.83) | 0.77 (0.67–0.86) | 0.69 (0.59–0.80) |
| | | M1 | 0.71 (0.68–0.73) | 0.73 (0.71–0.76) | 0.71 (0.67–0.74) | 0.68 (0.64–0.72) | 0.69 (0.63–0.74) | 0.66 (0.61–0.71) |
| Image DLS EfficientNet-V2-s [X2] | Two-Field Colour Fundal Photographs | R2+ \| M1 | 0.84 (0.82–0.86) | 0.82 (0.80–0.85) | 0.76 (0.74–0.79) | 0.85 (0.82–0.89) | 0.78 (0.73–0.84) | 0.77 (0.72–0.82) |
| | | p vs tabular DLS | <0.001 | <0.001 | 0.001 | <0.001 | 0.004 | 0.001 |
| | | R2+ | 0.96 (0.93–0.98) | 0.90 (0.85–0.95) | 0.82 (0.77–0.88) | 0.93 (0.90–0.97) | 0.90 (0.83–0.97) | 0.85 (0.74–0.97) |
| | | p vs tabular DLS | <0.001 | 0.002 | 0.005 | <0.001 | 0.038 | 0.069 |
| | | M1 | 0.83 (0.81–0.85) | 0.82 (0.79–0.85) | 0.76 (0.73–0.79) | 0.85 (0.81–0.89) | 0.78 (0.72–0.83) | 0.77 (0.72–0.82) |
| | | p vs tabular DLS | <0.001 | <0.001 | 0.003 | <0.001 | 0.008 | 0.001 |
| Multimodal DLS EfficientNet-V2-s [X2] & TabNet | Two-Field Colour Fundal Photographs & Risk Factor Characteristics[a] | R2+ \| M1 | 0.85 (0.83–0.87) | 0.85 (0.82–0.87) | 0.79 (0.76–0.82) | 0.85 (0.82–0.89) | 0.80 (0.76–0.85) | 0.79 (0.74–0.84) |
| | | p vs tabular DLS | <0.001 | <0.001 | <0.001 | <0.001 | <0.001 | <0.001 |
| | | p vs image DLS | 0.174 | 0.004 | 0.001 | 0.858 | 0.115 | 0.174 |
| | | R2+ | 0.95 (0.92–0.98) | 0.92 (0.87–0.96) | 0.85 (0.80–0.90) | 0.93 (0.88–0.97) | 0.93 (0.89–0.97) | 0.91 (0.84–0.98) |
| | | p vs tabular DLS | <0.001 | <0.001 | <0.001 | <0.001 | 0.001 | 0.001 |
| | | p vs image DLS | 0.546 | 0.431 | 0.159 | 0.322 | 0.123 | 0.067 |
| | | M1 | 0.84 (0.82–0.86) | 0.84 (0.82–0.87) | 0.79 (0.76–0.82) | 0.85 (0.80–0.89) | 0.79 (0.74–0.84) | 0.79 (0.74–0.83) |
| | | p vs tabular DLS | <0.001 | <0.001 | <0.001 | <0.001 | <0.001 | <0.001 |
| | | p vs image DLS | 0.137 | 0.003 | 0.003 | 0.999 | 0.247 | 0.258 |

Significant at $p < 0.017$, Bonferroni adjusted for the three predefined outcomes. Confidence intervals (95%) and significance were calculated using DeLong's method.

*AUROC* Area-under-the receiver operating characteristic, *CI* Confidence interval, *DLS* Deep learning system, R2+ | M1 Referable DR or maculopathy, *R2+* Referable DR, *M1* Referable maculopathy.

[a]Risk Factor characteristics=Age, sex, ethnicity, diabetes type, diabetes duration, best visual acuity, index of multiple deprivation rank.

Tabular DLS attributions generally indicated lower visual acuity, longer DM duration, younger age, male gender and Black, mixed or unspecified ethnicity increased DLS risk predictions for emergent referable disease (Supplementary Figs. S19, S20, S21, S22, S23 and S24). South Asian ethnicity and type I DM were variably associated with higher tabular DLS risk predictions, but socioeconomic deprivation had a limited effect on tabular DLS predictions.

## Discussion

We developed and validated tabular, image and multimodal DLS that predict future emergent referable DR, maculopathy or either within 1, 2 or 3 years, using data from two large, geographically distinct UK DESPs. Prognostic image and multimodal DLS demonstrated good performance and generalisation in the internal and external test populations for all prediction intervals and referable disease outcomes. The image and multimodal DLS significantly outperformed the risk factor characteristics based tabular DLS at all prediction intervals. The multimodal DLS, which incorporated risk factor characteristics and CFPs, showed moderate improvements in performance in internal and external test datasets compared to the image DLS, most notably for referable maculopathy prediction. Our study is unique in developing DLS that predict future emergent disease compared to the majority of prior approaches which trained DLS to detect DR at a present time point. To the best of our knowledge, our study is the first to develop and validate tabular, image and multimodal DLS that predicting emergent referable DR and referable maculopathy independently over 1, 2 and 3 year time horizons.

It is challenging to compare DLS performance between studies due to differences in study populations, retinal fields, risk-factor data, prediction intervals, DLS development, DR classification systems and predicted incident disease outcomes[22–24]. However, two relevant studies by Dai et al.[24] and Rom et al.[23] reported internal test emergent referable DR and maculopathy AUROC of 0.86 and 0.79 for the 1-year interval, 0.86 and 0.81 for the 2-year interval, and 0.85 and 0.78 for the 3-year interval. Only Dai et al. performed external validation with emergent referable DR and maculopathy AUROC ranging between 0.82 and 0.89 for the 1-year interval, 0.84−0.86 for the 2-year interval, and 0.82−0.83 for the 3-year interval[24]. These results closely

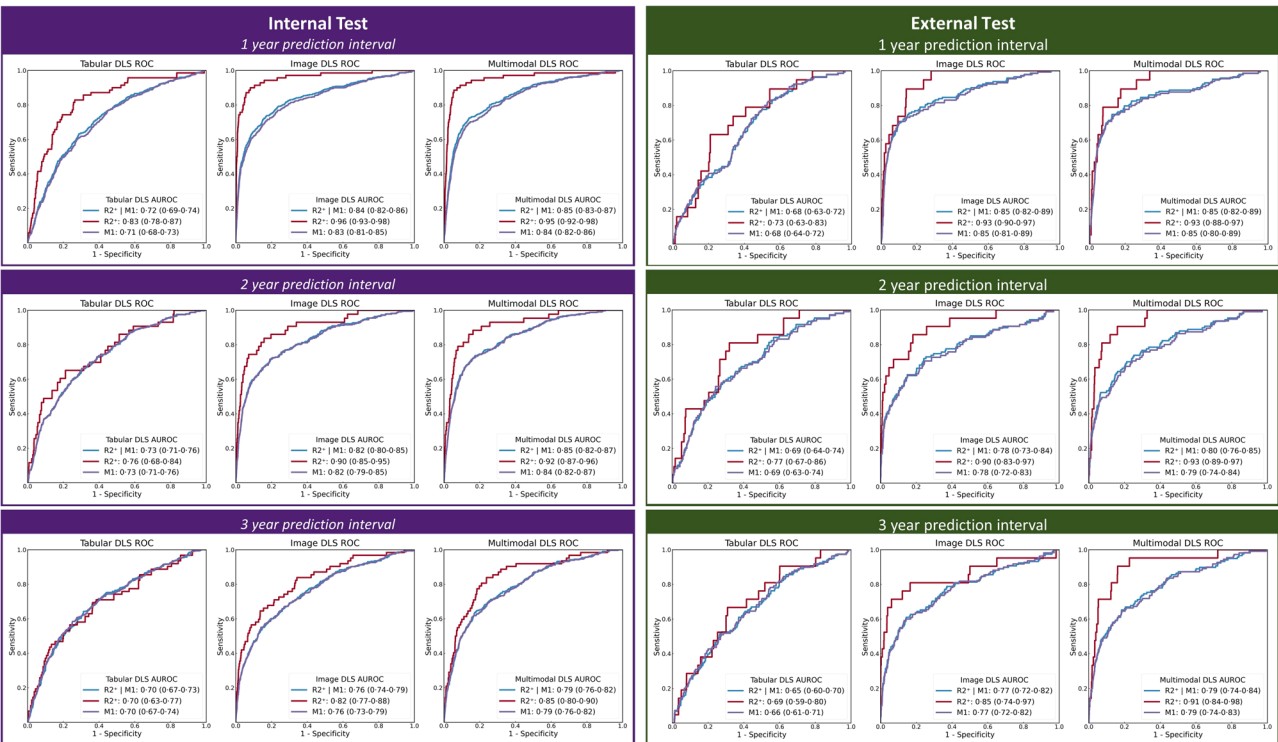

**Fig. 3 | Tabular, image and multimodal DLS receiver operating characteristic curves.** Tabular, image and multimodal DLS ROC curves for predicting progression to referable diabetic retinopathy (DR), maculopathy or either within 1, 2 or 3 years for the internal and external area-under-the receiver operating characteristic tests (AUROC) are summarised for each outcome. R2+ | M1=Referable DR or maculopathy. R2+=Referable DR. M1=Referable maculopathy. Confidence intervals (95%) are calculated using DeLong's method.

match the internal/external test performance of our multimodal DLS with emergent referable DR and maculopathy AUROC of 0.85/0.85 for the 1-year interval, 0.85/0.80 for the 2-year interval, and 0.79/0.79 for the 3-year interval, demonstrating the competitive performance of our approach. However, prior studies have not evaluated the use of DLS to independently predict emergent referable DR and referable maculopathy[12,13,17,18,21–24]. Since both referable DR and referable maculopathy have the potential to damage vision, albeit at varying rates, it is important that both outcomes are modelled and predicted independently[22–24]. Our DLS uses screening data from a single visit to predict both emergent referable DR and emergent referable maculopathy, allowing for selectable thresholds to meet a range of disease severity, sensitivity, and specificity requirements. Prior non-AI systemic risk factor based models also require individuals to undergo invasive blood tests which are not routinely performed by DR screening services[13]. In contrast, our DLS predicts emergent referable DR and maculopathy using non-invasive CFPs and clinicodemographic risk factor characteristics from a single screening visit, obviating the need for blood or other tests. Our study included the largest sample of eyes with known longitudinal outcomes used to develop our DLS than previously reported (110,837 unique eyes from 55,610 individuals (ours) vs 28,899 unique eyes[22], 17,190 individuals[24] and 19,531 individuals[21]). Our study development and test datasets were inclusive (<5% individuals excluded), diverse (≥40% non-white), and representative of the DESP population. Additionally, our datasets included a high degree of variation that is reflective of clinical practice, notably in regard to screening sites (27 SEL-DESP and 110 BSBC-DESP), imaging devices, and human graders. DLS prognostic performance generalised well to the external test population despite differences in ethnicity distributions, deprivation, and DM duration.

Undetected disease progression and subsequent visual loss are key concerns of individuals attending screening[9], therefore missing new onset proliferative DR, which can result in rapid visual decline if not urgently treated[31] is of particular clinical significance. Our DLS performed very well

in this regard as it was very rare that an eye progressed to proliferative DR undetected (0% external test, 0.3% internal test at 2-years) at the 80% sensitivity operating point. The DLS allows for the setting of independent operating points for emergent referable DR and emergent referable maculopathy to reduce the risk of missing incident proliferative disease. Individual-level performance, which simulated how eye-level predictions could be aggregated in a clinical setting, were near equivalent to eye-level results, and competitive to individual-level systemic-risk factor-driven models[12]. The rarity of emergent disease, in particular referable DR makes it challenging to draw robust conclusions for the subgroup analyses. Reduced DLS performance in the subgroup analyses should be considered conservative estimates due to the limited positive cases as a result of the low incidence of emergent referable disease.

We performed extensive ablation and attribution studies to identify CFP regions and risk factor characteristics which influenced DLS predictions. Image ablation studies showed that the central macula field was important to image DLS predictions of emergent referable disease. Uniquely, we found masking the nasal field did not significantly impact image DLS predictions. Variations in DLS performance noted with masking may be due to variations in the distribution of pertinent image features (which may vary by population) relative to the masked/unmasked image field area. Image attribution studies corroborated the ablation findings, demonstrating image DLS attention is concentrated near the fovea. This interesting finding suggests there may be prognostically important subclinical changes associated with future disease progression at the central macula. The majority of individuals attending screening have no clinically identifiable DR at baseline, and attribution analysis demonstrated no clinically discernible DR lesions in numerous cases (Supplementary Figs. S16−S18), therefore attribution findings are not wholly explainable by the presence of clinically detectable microaneurysms, exudates[5], or large retinal vessel calibre changes[32], as large vessels are absent at the fovea. However, changes may reflect capillary non-perfusion, non-clinically

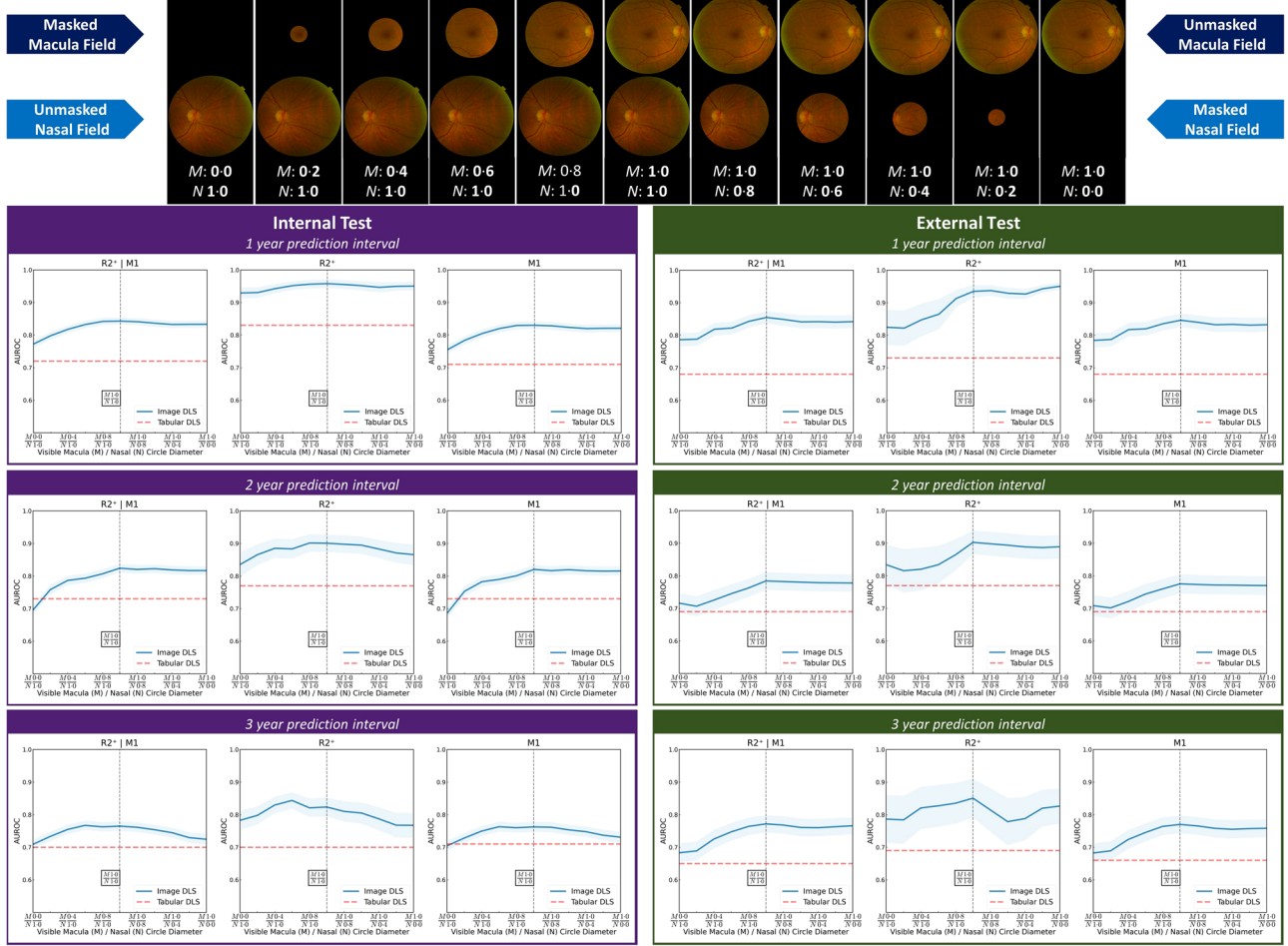

**Fig. 4 | Image DLS ablation.** Image ablation study demonstrating the effect of progressive macula or nasal field masking on predictive performance for referable diabetic retinopathy (DR), referable maculopathy or either outcomes at 1, 2 and 3 year intervals using a trained image DLS. The macula field is progressively unmasked with the nasal field held constant before the nasal field is progressively masked with the macula field held constant. Masking was applied during testing only. M (macula) and N (nasal) values indicate the diameter of a circle as a ratio of the width of the image within which the macula or nasal field is visible. A value of 1 represents no masking and 0 total masking. The AUROC was computed in 0.2 diameter steps, with 95% confidence intervals (light blue shaded area) computed using Delong's method. Tabular DLS AUROC are shown as reference (horizontal red dashed line) for each outcome and prediction interval, and the vertical black dashed line represents image DLS performance without any macula or nasal field masking.

discernible DR lesions or other foveal architectural changes which are detectable by the DLS within CFPs. Less specific patterns were evident on nasal field attributions, although the optic nerve head was consistently not highlighted as a significant feature. Potential explanations for nasal field attribution map observations are the occurrence of early DR lesions (e.g., microaneurysms)[15,16], large retinal vessels calibre changes[32] or altered vessel morphology, which are associated with DR progression. Overall, findings align with and support prior observations that the macula image contains important features that influence image DLS predictions of DR progression[21,22,24]. Tabular DLS attributions for emergent referable disease showed associations with longer DM duration and younger age, in agreement with the clinical and prognostic modelling literature[12,13,22,30]. Additionally, recognised DR progression risk factors of male gender, Black ethnicity and worse visual acuity were also associated with higher tabular DLS risk predictions[30]. Findings suggest that the tabular DLS learned clinically plausible associations between risk factor characteristics and their proportionate contribution to the risk of progression to referable disease.

A randomised-controlled trial established that individualised screening intervals could reduce DESP costs by 20%, appointments by over 40%, with non-inferior sight-threatening DR detection, and equivalent attendance[12]. With the UK DESP in the process of rolling out 2-yearly screening for low-risk individuals, DLS could be used as the risk-stratification engine that enables individualised risk-based screening. DLS use data from a single visit, and do not require invasive testing, therefore, they could be easier to deploy and scale to a national screening programme than the current best performing systemic risk factor based alternative[12]. DLS could be used to offer low-risk individuals a safe extension of follow-up intervals, reducing screening burden, saving time, reducing costs and increasing screening service capacity. Concurrently, high-risk individuals would benefit from timely referral for closer monitoring, and prompt sight-saving treatment. DLS could also be used internationally for prognostication, and the individualisation of DR screening which has been explored in Sweden and Iceland[10,18]. Additional applications for DLS include directing targeted preventative interventions, such as lifestyle modifications, for high-risk individuals, predicting other microvascular DM complications such as renal or cardiovascular disease[33,34], to guide management, identifying at-risk individuals for interventional clinical trials, and population-level forecasting to inform the planning of screening and hospital eye care services.

Study limitations are that self-reported data (e.g., ethnicity) used by the tabular DLS may be affected by bias and data entry errors. However, data entry errors are limited by the use of GP2DRS, an electronic system which directly transfers demographic data from general practice to the UK DESP. Additionally, grading variability may influence the derivation of emergent referable disease ground truth labels. However, grader error mainly affects

no DR vs mild DR determination, and not the classification of referable disease[35]. Furthermore, grading errors are minimised by the rigorous quality assurance, training and testing mechanisms embedded within the UK DESP (Supplementary Table S2). Significant care was taken to minimise bias during data curation and model development, including by setting judicious exclusion criteria to preserve dataset diversity and not unduly excluding individuals. We also used non-longitudinal data for pretraining instead of simply excluding all such data which meant even eyes without longitudinal data influenced DLS development. Finally we defined the incident disease ground truth labels using unaltered human grader DR and maculopathy grades to ensure DLS outputs would be applicable to current UK DR screening. However, residual bias, which can cause disparities in DLS performance between groups may remain because of differences in data acquisition, human grading, screening provision, or societal inequalities. Although there was a degree of variation in approved imaging devices between SEL-DESP and BSBC-DESP, image DLS generalisation to other retinal imaging devices requires validation. We did not evaluate DLS performance at shorter (<1 year) or longer (>3 years) intervals as routine screening intervals is normally between 1 and 3 year intervals, but this is a potential avenue for future research. Additionally, further experimentation with other base DLS (e.g., foundation models such as RETFound[36]), alternate DLS architectures (e.g., Transformers), DLS training techniques (e.g., unsupervised cross-sectional or longitudinal pre-training, DLS distillation, exponential moving averages), data augmentation (e.g., synthetic datasets), auxiliary tasks (e.g., co-detection of incident disease risk factors such as age, sex, visual acuity and diabetes duration) and predictive approaches (survival-based DLS prediction) warrant further exploration. Additionally, future studies could explore how temporal changes in retinal images and risk factors correlate with DLS predictions. Larger validation populations would also allow more robust conclusions on the performance of the prognostic DLS, including within demographic subgroups. Future research should also prospectively assess the safety and efficacy of the DLS, as well as other important aspects of model implementation including fairness, cost, acceptability, human-DLS interactions, and integration within screening and healthcare pathways. Finally, the determination of operating thresholds with clinically appropriate sensitivity and specificity that meet the needs of people with diabetes and screening services will need to be determined[8,9,12].

In conclusion, we developed and validated tabular, image and multimodal DLS that accurately predict 1, 2 or 3 year emergent referable DR and referable maculopathy. DLS could be used to enable individualised risk-based diabetic eye screening, and to detect at-risk individuals to inform earlier preventive interventions.

## Data availability

De-identified data used for development and internal testing are not publicly available at present due to the absence of authorisation for public data sharing. However, the data used for external validation can be requested from HDRUK (INSIGHT) with access/approval dependant on meeting data governance and application criteria. The source data for Fig. 1 and Fig. 2 can be found in Supplementary Data 1. The source data for Fig. 3 and Fig. 4 (unmasked AUROC and tabular DLS points) can be found in Table 1.

## Code availability

Open-source resources and code were used to develop and validate the prognostic deep learning systems including python, pytorch, timm, and TabNet. Image curation DLS code and test data from a separate open access dataset and respective test labels are all publicly available at https://github.com/pnderitu/DUK_Automated_Curation[37].

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

## Acknowledgements
The study external validation data and compute was supported by INSIGHT, the Health Data Research Hub in Eye disease and Oculomics, which is affiliated with Health Data Research UK. The model development study was funded by Diabetes UK via a Sir George Alberti research training fellowship grant to Dr Paul Nderitu (20/0006144) who was supervised by Professor Tim Jackson and Dr Christos Bergeles. The external validation study was funded by King's College Hospital Charity via a research grant to Dr Paul Nderitu and Professor Tim Jackson (D2312/102022/Jackson/991). The study was also supported by a Wellcome/EPSRC Centre for Medical Engineering grant (WT 203148/Z/16/Z) to Dr Christos Bergeles. Study funders did not have access to the study data, nor did they influence the study design, DLS development, data analysis, or manuscript preparation. The corresponding author had the final responsibility for the decision to submit the manuscript for publication.

## Author contributions
P.N., J.M.N., L.W., S.M., D.H., M.J.C., M.M., C.B. and T.J. formulated the study concept. P.N., J.M.N., M.J.C., C.B. and T.J. were involved in the study design. P.N., J.M.N., C.B. and T.J. created the study protocol. P.N. and L.W. performed the data extraction. P.N. performed grading used to develop the curation D.L.S. P.N. and J.N. performed the coding & model development. P.N., J.M.N., S.M., M.J.C., C.B. and T.J. performed the primary data analysis. J.M.N., S.M., M.J.C., C.B. and T.J. provided study oversight and supervision. P.N. drafted the initial manuscript. Manuscript revision and final approval was performed by P.N., J.M.N., L.W., S.M., D.H., M.J.C., M.M., C.B. and T.J.

## Competing interests
P.N., J.M.N., L.W., D.H., M.J.C., M.M. and C.B. have no conflicts of interest to declare. S.M. has received speaker and advisory board fees from Bayer and Allergan and has received research grants from Novartis. T.J. has received payment as a clinical expert by solicitors acting for REGENERON. T.J. employer (King's College Hospital) receives funding for participants enrolled on commercial clinical trials of diabetic retinopathy including THR149-002 (OXURION), NEON NPDR (BAYER), RHONE-X (ROCHE) and ALTIMETER (ROCHE).
