## [Peer Review File · Communications Medicine]

Reviewers' comments:

Reviewer #1 (Remarks to the Author):

The study addresses a clinically significant problem by aiming to enable individualized screening for diabetic retinopathy, which could lead to more efficient and effective management of the condition. The use of deep learning for predicting the progression of DR and maculopathy is a novel approach that leverages recent advances in artificial intelligence.

The results are promising, with the multimodal DLS showing good performance and generalization in both internal and external test populations. The ability of the DLS to predict emergent referable DR and maculopathy with high accuracy is a significant achievement that could have a substantial impact on patient care.

The manuscript should discuss how its findings compare with those of the recent paper published in Nature Medicine by Dai et al. (2024), titled "A deep learning system for predicting time to progression of diabetic retinopathy." Both studies aim to predict the progression of DR using deep learning, but there may be differences in methodologies, datasets, and outcomes that are worth highlighting.

The authors have acknowledged some limitations, such as the potential for bias in self-reported data and the influence of grading variability. Future work could include prospective validation of the DLS, exploration of shorter or longer prediction intervals, and assessment of the DLS's utility in different clinical settings.

The authors could perform a subgroup analysis to explore the performance of the deep learning systems (DLS) in different demographic groups (e.g., age, sex, ethnicity) or clinical subgroups (e.g., type of diabetes, duration of diabetes, baseline retinopathy grade). This could help identify any disparities in the predictive performance of the DLS and provide insights into its generalizability across diverse populations.

The authors could perform a sensitivity analysis to assess the robustness of their results to different thresholds for defining referable diabetic retinopathy and maculopathy. This would help understand how variations in the definition of referable disease might affect the predictive performance of the DLS.

If longitudinal data are available, the authors could explore the temporal changes in retinal images and risk factors over time and how these changes correlate with the progression of diabetic retinopathy and maculopathy. This analysis could provide insights into the natural history of the disease and the predictive value of longitudinal data.

The authors could conduct a feature importance analysis to identify the most predictive features used by the tabular DLS. This could provide insights into the risk factors that are most strongly associated with the progression of diabetic retinopathy and maculopathy.

If possible, the authors could validate their DLS in different geographic locations or healthcare settings to assess its generalizability and performance in diverse populations and environments.

Reviewer #2 (Remarks to the Author):

This is an interesting paper using DL technique to estimate risk of diabetic retinopathy progression to allow some customisation of screening intervals. Some of the methods need clarification and possible expanding and there are limitation the authors do no acknowledge

There are other recent risk stratification paper and DL papers that have not been cited e.g. Olvera-Barrios at el Ethnic disparities in progression rates for sight-threatening diabetic retinopathy in diabetic eye screening: a population-based retrospective cohort study - BMJ Open Diabetes Res Care 2023

Lines 120-122; 183 – believe the authors have muddled specificity with sensitivity.

The authors pre-trained their model on an internal dataset with a limited number of images (~50k). It would be interesting to see the impact of extending this pre-training with more images sourced from public datasets with relevant labels. Justification on the number of images used for pre-training should be provided.

The authors compare results across their own implemented deep learning systems (DLS) effectively. Further discussion on how their performance compares to other DLS would be helpful to include to give an indication of comparative performances. It is recognized that direct comparison is difficult given the application to different datasets and the focus on other studies to predict progression to referable DR without investigation of the prediction of emergent referable maculopathy independently, but a discussion to acknowledge this aspect would be a useful inclusion in the paper.

The auxiliary objective of detecting mild to moderate DR at baseline improves results and is a good addition to incorporate the learnt features. It may be helpful to include in the discussion any further auxiliary tasks that could be considered.

EfficientNetV2 and TabNet have been incorporated in the DLS and some reasons for their selection are provided within the supplementary material. Further comment to indicate suitability of these approaches in future systems where training requirements may be higher would be a helpful addition. As far as I can see, computational resource used for the implementation has not been provided and should be included. The DLS takes the mean of the tab and image data outputs, but justification for this approach is unclear. Further discussion could provide clarity here to understand options for combining the outputs earlier in the system or in an alternative manner.

The attribution maps shown lack detail due to the assumed alignment as we would expect. Could further detail be provided by providing images of individual attribution maps as examples to enhance the visualization of key areas?

Line 235 mentions a link between dataset diversity, using non-longitudinal data for pre-training and definition of ground truths. Further clarification of this concept should be provided here to improve clarity for the reader.

In Figure 4. showing image ablation, some AUROC values vary across the internal and external test and in some cases the masked AUROC is higher than the unmasked. Further discussion relating to this figure and its results should be provided.

Authors refer to performance by stratification of demographic groups, but no data are presented. Referral DR is the primary health outcome for these analyses, but with less than a maximum of 70 cases in the internal test set and only 22 cases of referable DR in the external data set is not sufficient to provide evidence of “preserved performance” by for example age group or ethnicity. As

such the study is under powered here to answer these questions and is major limitation of the study.

Reviewer #3 (Remarks to the Author):

SUMMARY:

The authors developed deep learning systems (DLS) to predict the progression of referable DR, maculopathy or either within 1-3 years. The study evaluated the performance of tabular DLS, image DLS, and multimodal DLS using risk factors and colour fundal photographs. The study is based on datasets from two geographical locations for internal and external validation, demonstrating the accurate prognostic prediction performance. However, there are still some issues and considerations that need further discussion and clarification:

Major comments:

1. Experimental analysis.

- The authors utilized the 80% sensitivity operating point for evaluating the performance of the DLS. It would be beneficial for the authors to provide an explanation for their choice of using 80% sensitivity. As we know, in screening scenarios, usually higher sensitivity is desired. It is suggested to conduct a more comprehensive analysis of the models' performance at different thresholds.

- What is the reason that the multimodal model yield lower prognostic results for the 1-year indicator compared to the pure image model's 1-year prognostic results? The authors are suggested to provide a possible explanation for this discrepancy.

- The authors have successfully developed and validated DLS for predicting progression for the 1, 2, and 3- year prediction intervals. The prediction of progression within a time period is also a critical problem [1][2], such as progression within 3 years. It would be interesting to know whether the developed DLS model is capable of assessing a patient's risk within a specific time period.

[1] Arcadu, F., Benmansour, F., Maunz, A. et al. Deep learning algorithm predicts diabetic retinopathy progression in individual patients. *npj Digit. Med.* 2, 92 (2019).

[2] Bora, Ashish et al. "Predicting the risk of developing diabetic retinopathy using deep learning." The Lancet. Digital health vol. 3,1 (2021): e10-e19.

2. Baseline characteristics and datasets

- The authors used risk factor characteristics alone (tabular DLS), two-field CFPs alone (image DLS) or both for the development of AI model. Does the model incorporate any non-invasive risk factors such as hypertension, triglyceride, low-density lipoprotein cholesterol levels, or renal disease?

3. Methodology

-The EfficientNet-V2 and TabNet models were employed for the analysis of image and tabular data. However, the selection of these models is based on their performance in other related tasks, and the article lacks a sufficient explanation as to why these models are particularly suited for the task at hand, as well as whether other model configurations were attempted.

-The methodology lacks a comparison with other baseline methods, which could provide further validation of the proposed approach's effectiveness.

- The use of two feature extraction networks (EfficientNetV2) that do not share parameters to process multi-view images lacks clear support. This approach could potentially overlook the interrelationships and consistency between different eye images, which is important for disease diagnosis.

Minor comments:

- The authors employed RMSProp as the optimization method for pretraining, whereas used AdamW for longitudinal training. Could you provide some insights or references explaining the reason of this setting for the benefit of readers?

- Annotations could be added to subplots a and b in Table 1 to better illustrate the process of constructing the dataset.

- Line 88: There appears to be a missing closing parenthesis. Line 121: there is an extraneous 's' present.

Response To Reviewer Comments

Reviewer 1

Summary: The study addresses a clinically significant problem by aiming to enable individualized screening for diabetic retinopathy, which could lead to more efficient and effective management of the condition. The use of deep learning for predicting the progression of DR and maculopathy is a novel approach that leverages recent advances in artificial intelligence. The results are promising, with the multimodal DLS showing good performance and generalization in both internal and external test populations. The ability of the DLS to predict emergent referable DR and maculopathy with high accuracy is a significant achievement that could have a substantial impact on patient care.

#	Comment & Response	Changes in Manuscript
R1.1	Comment: The manuscript should discuss how its findings compare with those of the recent paper published in Nature Medicine by Dai et al. (2024), titled "A deep learning system for predicting time to progression of diabetic retinopathy." Both studies aim to predict the progression of DR using deep learning, but there may be differences in methodologies, datasets, and outcomes that are worth highlighting. Response: We thank the reviewer for this useful suggestion. Although there are important differences in populations, modelling approaches, imaged fields and DR classification systems, we agree that a comparison with the Dai paper, as well as another recent paper (Rom, Y., Aviv, R., Ianchulev, T. & Dvey-Aharon, Z. Predicting the future development of diabetic retinopathy using a deep learning algorithm for the analysis of non-invasive retinal imaging. BMJ Open Ophthalmology 7, e001140, doi:10.1136/bmjophth-2022-001140) is worthwhile, and will provide the reader with additional context of how our DLS performance fits into the wider literature. We have added this detail to the manuscript discussion: 'It is challenging to compare DLS performance between studies due to differences in study populations, retinal fields, risk-factor data, prediction intervals, DLS development, DR classification systems and predicted incident disease outcomes.22-24 However, two relevant studies by Dai et al24 and Rom et al23 reported internal test emergent referable DR and maculopathy AUROC of 0.86 and 0.79 for the 1-year interval, 0.86 and 0.81 for the 2-year interval, and 0.85 and 0.78 for the 3-year interval. Only Dai et al performed external validation with emergent referable DR and maculopathy AUROC ranging between 0.82-0.89 for the 1-year interval, 0.84-0.86 for the 2-year interval, and 0.82-0.83 for the 3-year interval.24 These results closely match the internal/external test performance of our multimodal DLS with emergent referable DR and maculopathy AUROC of 0.85/0.85 for the 1-year interval,	Main text: P7 line 164-173

	0.85/0.80 for the 2-year interval, and 0.79/0.79 for the 3-year interval, demonstrating the competitive performance of our approach.'	
R1.2	Comment: The authors have acknowledged some limitations, such as the potential for bias in self-reported data and the influence of grading variability. Future work could include prospective validation of the DLS, exploration of shorter or longer prediction intervals, and assessment of the DLS's utility in different clinical settings.	
	Response: We agree and have raised the above points in the manuscript discussion: 'We did not evaluate DLS performance at shorter (<1 year) or longer (>3 years) intervals as routine screening intervals is normally between 1 and 3 year intervals, but this is a potential avenue for future research.'	Main text: P10-11 line 259-260
R1.3	Comment: The authors could perform a subgroup analysis to explore the performance of the deep learning systems (DLS) in different demographic groups (e.g., age, sex, ethnicity) or clinical subgroups (e.g., type of diabetes, duration of diabetes, baseline retinopathy grade). This could help identify any disparities in the predictive performance of the DLS and provide insights into its generalizability across diverse populations.	
	Response: We include detailed subgroup analysis with stratification by age group, sex and ethnicity in the supplementary materials and show this for tabular, image and multimodal DLS for all prediction intervals and outcomes; please see supplementary table S10, S11 and S12. To make this information more visible to readers we have highlighted reference to these tables in the results section of the manuscript: 'Subgroup analysis whereby DLS performance was stratified by age, sex and ethnicity were performed, with results shown in supplementary table S10-S12.' We felt that subgroup analysis by these protected characteristics were the most important subgroups to add to the manuscript. Additional stratification by clinical subgroups and other variables would result in a very large subgroup analysis due to the number of DLS, prediction intervals and outcomes which may detract from the protected characteristics subgroup analysis, hence we have not included this but it remains an avenue of future exploration.	Main text: P6 line 132-133
R1.4	Comment: The authors could perform a sensitivity analysis to assess the robustness of their results to different thresholds for defining referable diabetic retinopathy and maculopathy. This would help understand how variations in the definition of referable disease might affect the predictive performance of the DLS.	
	Response: We provide sensitivity analysis in three forms within the supplementary results. Firstly, we compute the full confusion matrix for all DLS and intervals at a threshold of 80% sensitivity to match an RCT from the UK, which examined individualised screening using a risk-prediction model at the same sensitivity threshold (Broadbent, D. M. et al. Safety and cost-effectiveness of individualised screening for diabetic retinopathy: the ISDR open-label, equivalence RCT. Diabetologia 64, 56-69, doi:10.1007/s00125-020-05313-2, 2021). The results are shown in supplementary table S7. Secondly, we also computed how the false negative rate changes as a function of threshold, as shown in	Main text: P5-6 line 120-136. Supplementary: tables S7, S8, S9

	supplementary figure S8, as well as how the positive and negative prediction values also change with thresholds as shown in supplementary figure S9. These provide the reader with data on how changes in thresholds may affect the models clinical performance. These data are provided to readers within the following text in the manuscript: Results section, paragraph 4, line 120-132 The definition of emergent referable DR and maculopathy, which is our ground truth, is fixed as it is based on the human graders' classification and on UK NSC DR grading criteria; these definitions are not modifiable post-hoc. The ground truth uses the current DR screening definitions as the DLS would be used within the current pathways.	
R1.5	Comment: If longitudinal data are available, the authors could explore the temporal changes in retinal images and risk factors over time and how these changes correlate with the progression of diabetic retinopathy and maculopathy. This analysis could provide insights into the natural history of the disease and the predictive value of longitudinal data.	
	Response: We thank the reviewer for the helpful suggestion. Unfortunately, the frame from which we train our DLS uses the final visit as the prediction visit, hence we don't have additional longitudinal data from this point onwards to assess if the DLS outputs are predictive of longer term DM / DR outcomes (akin to Bora et al's paper) although this is an area of interest in future research and exploration in planned prospective evaluations. We have added the following to the discussion: 'Additionally, future studies could explore how temporal changes in retinal images and risk factors correlate with DLS predictions'	Main text: P11 line 266-267
R1.6	Comment: The authors could conduct a feature importance analysis to identify the most predictive features used by the tabular DLS. This could provide insights into the risk factors that are most strongly associated with the progression of diabetic retinopathy and maculopathy.	
	Response: Tabular DLS attribution analysis are provided in the supplementary figures S19, S20 and S21 for all prediction intervals for both internal and external validation sets. We present the tabular DLS attribution findings in the main manuscript results and discussion and show that clinically plausible and proportional risk attributions are learned by the tabular DLS. These data are provided to readers within the following text in the manuscript: Results section, final paragraph, line 147-150	
R1.7	Comment: If possible, the authors could validate their DLS in different geographic locations or healthcare settings to assess its generalizability and performance in diverse populations and environments.	
	Response: The interval and external test set populations used in this study came from two geographically separate screening services. Within each, there are multiple sites. Hopefully this goes some way to addressing the reviewer's helpful suggestion. We acknowledge the need for additional validation in the limitations as follows:	Main text: P11 line 267-268

	'Larger validation populations would also allow more robust conclusions on the performance of the prognostic DLS, especially within smaller demographic subgroups'	
--	---	--

Reviewer 2:

Summary: This is an interesting paper using DL technique to estimate risk of diabetic retinopathy progression to allow some customisation of screening intervals. Some of the methods need clarification and possible expanding and there are limitations the authors do no acknowledge.

#	Comment & Response	Changes in Manuscript
R2.1	Comment: There are other recent risk stratification paper and DL papers that have not been cited e.g. Olvera-Barrios at el Ethnic disparities in progression rates for sight-threatening diabetic retinopathy in diabetic eye screening: a population-based retrospective cohort study - BMJ Open Diabetes Res Care 2023 Response: We thank the reviewer for this comment. The Olvera-Barrios paper applies a risk stratification rule being proposed for use in UK DR screening, namely if no DR in two consecutive visits then the patient is low risk. The rule bases system is derived from Stratton et al (A simple risk stratification for time to development of sight-threatening diabetic retinopathy. Diabetes Care 36, 580-585, 2013), and validated by Lesse et al (Progression of diabetes retinal status within community screening programs and potential implications for screening intervals. Diabetes Care 38, 488-494, 2015). We reference these papers for this rule-based risk-stratification systems in the introduction (page 4, line 74-79). We agree that, importantly, Olvera-Barrios et al demonstrate potential issues with rule-based risk stratification systems with respect to a potentially higher-risk of missing sight-threatening DR in different ethnic groups. We make this point and reference Barrios et al in the introduction (page 4, line 78-80) but have made this reference clearer in the main text and referenced the author explicitly: 'A recent study by Olvera-Barrios et al also found that retinopathy-grade based risk-stratification approaches could delay the detection of sight-threatening DR, particularly for patients of black ethnic origin.'	Main text: P4 line 78-80
R2.2	Comment: Lines 120-122; 183 – believe the authors have muddled specificity with sensitivity. Response: We thank the reviewer for this comment as we agree the current text is confusing. We define sensitivity as the DLS ability to correctly discriminate if an eye will develop incident referable disease between the baseline and predict visit for the given time interval (high-risk) from all eyes who had this outcome. The specificity is the DLS ability to correctly distinguish eyes that will not develop referable disease in the given time frame (low-risk) from all non-progressing eyes. In the threshold analysis, we set the sensitivity of the model to 80% and then evaluate the DLS specificity, namely its performance in detect eyes at low-risk of incident referable disease.	Main text: P5-6, line 120-124

	The current text is confusing as it states, ‘At 80% sensitivity, multimodal DLS internal/external test specificity for predicting emergent referable DR, maculopathy or either ...’ which seems to refer to the high risk group w.r.t to specificity. Therefore, we modified the sentence to: ‘At 80% sensitivity, multimodal DLS internal/external test specificity for detecting eyes at low-risk of emergent referable DR, maculopathy or either ...’	
R2.3	Comment: The authors pre-trained their model on an internal dataset with a limited number of images (~50k). It would be interesting to see the impact of extending this pre-training with more images sourced from public datasets with relevant labels. Justification on the number of images used for pre-training should be provided. Response: We thank the reviewer for this insightful suggestion. This had been a consideration during image DLS development. However, differences between UK DR screening grading to the international DR classification systems, and the lack of sufficient metadata descriptions of age, sex and ethnicity distributions in publicly available DR datasets deterred us from using these data for supervised pre-training. Additionally, the sizable public datasets are predominantly single-field (EyePACS/Kaggle and DDR) and not 2-field as we required for this study. However, pre-training is an area that could be developed further in the future, perhaps with unsupervised approaches or using a foundation model as our base model (e.g. RefFound). We have added this comment to the end of the discussion: ‘Additionally, further experimentation with other base DLS (e.g. foundation models such as RETFound), alternate DLS architectures (e.g. Transformers), DLS training techniques (e.g. unsupervised cross-sectional or longitudinal pre-training, DLS distillation, exponential moving averages), data augmentation (e.g. synthetic datasets), auxiliary tasks (e.g. co-detection of incident disease risk factors such as age, sex, visual acuity and diabetes duration) and predictive approaches (survival-based DLS prediction) warrant further exploration.’	Main text: P10-11, line 261-265
R2.4	Comment: The authors compare results across their own implemented deep learning systems (DLS) effectively. Further discussion on how their performance compares to other DLS would be helpful to include to give an indication of comparative performances. It is recognized that direct comparison is difficult given the application to different datasets and the focus on other studies to predict progression to referable DR without investigation of the prediction of emergent referable maculopathy independently, but a discussion to acknowledge this aspect would be a useful inclusion in the paper. Response: Thank you for the comment which was also made by Reviewer #1. We signpost you to our response R1.1 whereby we discuss how we have added these comparisons to the discussion for the composite incident referable DR and maculopathy predictions and compare recent papers by Dai et al (2024) and Rom et al (2022).	Main text: P7 line 164-173
R2.5	Comment: The auxiliary objective of detecting mild to moderate DR at baseline improves results and is a good addition to incorporate the learnt features. It may be helpful to include in the discussion any further auxiliary tasks that could be considered.	

	Response: We theorise that detecting other risk factors of disease progression (as shown in the tabular DLS attributions such as age, sex, and diabetes duration) may be useful auxiliary tasks that could improve prognostic DLS performance, as identifying retinal features for these risk factors are likely to also correlate with the incident disease prediction task. We have added detail to the main text on potential on the exploration of auxiliary tasks as an avenue for future reseearch as follows: 'Additionally, further experimentation with other base DLS (e.g. foundation models such as RETFound), alternate DLS architectures (e.g. Transformers), DLS training techniques (e.g. unsupervised cross-sectional or longitudinal pre-training, DLS distillation, exponential moving averages), data augmentation (e.g. synthetic datasets), auxiliary tasks (e.g. co-detection of incident disease risk factors such as age, sex, visual acuity and diabetes duration) and predictive approaches (survival-based DLS prediction) warrant further exploration.' As a note, auxiliary tasks related to DR progression prediction were well described by Bora et al (Predicting the risk of developing diabetic retinopathy using deep learning. Lancet Digit Health. 2021 Jan;3(1):e10-e19) and included cross-sectional classification of DR severity, predicting incident disease at various severities and time points. This paper inspired the design of some of the auxiliary tasks and hence we have additionally referenced the study in the methods as follows: 'Additionally, cross sectional detection of DR severity has been reported to improve DLS performance in predicting DR progression.^{22'}	Main text: P10 line 261-265 and P13 line 334-335
R2.6	Comment: EfficientNetV2 and TabNet have been incorporated in the DLS and some reasons for their selection are provided within the supplementary material. Further comment to indicate suitability of these approaches in future systems where training requirements may be higher would be a helpful addition. As far as I can see, computational resource used for the implementation has not been provided and should be included. Response: We used Efficient Net V2 / TabNet as our base models during development, because they were the best performing models in the literature within their parameter envelope. However, with increasing computational resources and newer approaches, we agree that other models (hybrid models, Transformers, foundation models i.e. RETFound, vision language ...) are worth exploring, in addition to a number of other DLS development techniques. We have added details to elaborate on this further in the discussion, as areas for future exploration: 'Additionally, further experimentation with other base DLS (e.g. foundation models such as RETFound³¹), alternate DLS architectures (e.g. Transformers), DLS training techniques (e.g. unsupervised cross-sectional or longitudinal pre-training, DLS distillation, exponential moving averages), data augmentation (e.g. synthetic datasets), auxiliary tasks (e.g. co-detection of incident disease risk factors such as age, sex, visual acuity and	Main text: P10 line 261-265

	diabetes duration) and predictive approaches (survival-based DLS prediction) warrant further exploration.' On the second comment about the computational resources used, we have added detail about this in the supplementary materials as follows: 'A Windows 10 Pro workstation with a 32 core Intel Xenon CPU, 128GB of RAM and x2 24GB Quadra RTX P6000 GPUs was used for DLS development and internal testing. External testing was performed on a cloud-based trusted research environment with a GPU with 12GB of RAM.'	Supplementary: P4, line 123-125
R2.7	Comment: The DLS takes the mean of the tab and image data outputs, but justification for this approach is unclear. Further discussion could provide clarity here to understand options for combining the outputs earlier in the system or in an alternative manner.	
	Response: We reference our reasoning in the Methods / Model development section and reference this as follows (P13 line 337-340): 'The multimodal DLS was created from the ensemble of the trained image and tabular DLS by computing the mean of their predictions at test time as early experiments showed this strategy was superior to end-to-end training of a single multimodal DLS by concatenating the final feature vectors of the image and tabular DLS (supplementary methods pp 6).' Additionally, on P6 of the Supplementary methods/ Multimodal DLS section we discuss that prediction level fusion (mean of predictions) was better than end-to-end training with late fusion (concatenate the final feature vector of the image and tabular DLS) (AUROC +0.01-0.04) when we performed the experiments on 2-year prediction and tested in the internal dataset. We have made this section clearer in the supplementary and npw reads as follows: 'Experiments showed that prediction-level fusion performed better than end-to-end training with late fusion defined as the concatenation of the final feature vector of the image and tabular DLS. The 2-year internal test set multimodal DLS AUROC for prediction-level fusion vs late fusion end-to-end training were 0.92 (95% CI: 0.87-0.96) vs 0.88 (0.82-0.94), 0.84 (0.82-0.87) vs 0.83 (0.81-0.86) and 0.85 (0.82-0.87) vs 0.83 (0.81-0.86) for predicting emergent referable DR, maculopathy or either respectively. On the basis of this findings, prediction-level fusion was used for all multimodal DLS.'	
R2.8	Comment: The attribution maps shown lack detail due to the assumed alignment as we would expect. Could further detail be provided by providing images of individual attribution maps as examples to enhance the visualization of key areas?	
	Response: We thank the reviewer for this excellent suggestion. We have added new supplementary figures (S16-18) showing attributions for three individual eye samples with incident referable disease, one for each of the 1, 2 and 3 year prediction intervals, to demonstrate better localisation of pertinent image	Supplementary: Fig S16- S18

	DLS features in support of the mean attribution maps, with additional captions. We reference the figure on P6 of the results as follows: 'Attributions for sample eyes with incident disease are also provided in supplementary figures S16-18 for better localisation of pertinent image features.' We also reference them again on P9 of the discussion as follows: 'The majority of individuals attending screening have no clinically identifiable DR at baseline, and attribution analysis demonstrated no clinically discernible DR lesions in numerous cases (supplementary figures S16-S18)'	Main text: P6 line 144-145 Main text: P9 line 211-213
R2.9	Comment: Line 235 mentions a link between dataset diversity, using non-longitudinal data for pre-training and definition of ground truths. Further clarification of this concept should be provided here to improve clarity for the reader. Response: We apologise for any confusion, but this sentence was meant to demonstrate the design choices made to minimise undue biases during DLS development. Namely, we did not exclude eyes and data without longitudinal follow-up, but instead used them for pre-training, ensuring all eyes provided some contribution to DLS development. We wanted to stress that we maximised dataset diversity by not excluding participants wherever possible, and used judicious exclusion criteria to avoid over-sanitising the dataset. Finally, we keep the human grader labels when constructing the incident disease ground-truth, and did not create a modified definition to ensure DLS outputs would be applicable to UK DR screening. We have clarified the above by reconfiguring the sentence in the main text accordingly: 'Significant care was taken to minimise bias during data curation and model development, including by setting judicious exclusion criteria to preserve dataset diversity and not unduly excluding individuals. We also used non-longitudinal data for pretraining instead of simply excluding all such data which meant even eyes without longitudinal data influenced DLS development. Finally, we defined the incident disease ground truth labels using unaltered human grader DR and maculopathy grades to ensure DLS outputs would be applicable to current UK DR screening.'	Main text: P10 line 250-255
R2.10	Comment: In Figure 4. showing image ablation, some AUROC values vary across the internal and external test and in some cases the masked AUROC is higher than the unmasked. Further discussion relating to this figure and its results should be provided. Response: We agree that there is variance in image DLS performance, that is occasionally higher in a partially masked image vs completely unmasked images. We hypothesise that there are pertinent image features in each field which influence DLS predictions. Depending on the masked or unmasked image field area and its interaction with the distribution of these image features which may differ between the internal and external test populations could explain the fluctuation in DLS performance, and result in the non-linear relationship between masking and DLS performance. Indeed, the choice of masking could also be varied (e.g. circular, square etc) and this could also influence the	Main text: P9 line 207 -209

	observed performance patterns. We have added qualification within the discussion to add that the variations in ablation could occur with masking, as it is dependant on the distribution of the pertinent image features: 'Variations in DLS performance noted with masking may be due to variations in the distribution of pertinent image features (which may vary by population) relative to the masked/unmasked image field area.'	
R2.11	Comment: Authors refer to performance by stratification of demographic groups, but no data are presented.	
	Response: The data are provided within the supplementary tables S10-12 and we signpost this in the main text (Page 4, line 132-133) and referenced as follows: 'Subgroup analysis whereby DLS performance was stratified by age, sex and ethnicity were performed, with results shown in supplementary table S10-S12.'	
R2.12	Comment: Referral DR is the primary health outcome for these analyses, but with less than a maximum of 70 cases in the internal test set and only 22 cases of referable DR in the external data set is not sufficient to provide evidence of "preserved performance" by for example age group or ethnicity. As such the study is under powered here to answer these questions and is major limitation of the study.	
	Response: Thank you for this useful comment and we do agree the rarity of incident referable DR cases means it is not possible to robustly comment on generalisation by subgroup (especially smaller subgroups). We have amended the discussion to temper our conclusions in this respect as follows: 'Subgroup analyses suggested DLS had good performance for larger subgroups, but the rarity of emergent referable DR and maculopathy means it is not possible to draw robust conclusions for smaller subgroups.' and added this point to the limitations: 'Larger validation populations would also allow more robust conclusions on the performance of the prognostic DLS, especially within smaller demographic subgroups.'	Main text: P8 line 198 -200 and P11 line 267 -268

Reviewer 3

Summary: The authors developed deep learning systems (DLS) to predict the progression of referable DR, maculopathy or either within 1-3 years. The study evaluated the performance of tabular DLS, image DLS, and multimodal DLS using risk factors and colour fundal photographs. The study is based on datasets from two geographical locations for internal and external validation, demonstrating the accurate prognostic prediction performance. However, there are still some issues and considerations that need further discussion and clarification:

#	Comment & Response	Changes in Manuscript
R3.1	Comment: The authors utilized the 80% sensitivity operating point for evaluating the performance of the DLS. It would be beneficial for the authors to provide an explanation for their choice of using 80% sensitivity. As we know, in screening scenarios, usually higher sensitivity is desired. It is suggested to conduct a more comprehensive analysis of the models' performance at different thresholds.	
	Response: In this study, the DLS are used prognostically for future disease prediction, which differs to cross-sectional disease detection as is required for DR screening, and in which the desirable characteristics are a sensitivity of ~80% and specificity ~95% (Scanlon PH. Acta Diabetol 2017. 54(6):515-525). A prior randomised controlled trial in the UK (Eleuteri, A. et al. Individualised variable-interval risk-based screening for sight-threatening diabetic retinopathy: the Liverpool Risk Calculation Engine. Diabetologia 60, 2174-2182, doi:10.1007/s00125-017-4386-0, 2017) used a systemic risk factor model to assess its application to individualised DR screening. In this RCT, they showed at 80% sensitivity, they had non-inferior referable DR detection and equivalence in attendance to screening for high vs low risk patients. This is why we chose an 80% sensitivity as our threshold, to both compare to this systemic risk factor model, but also this is the best evidence for what threshold should be used for a prognostic DLS in the UK DR screening. We reference the study and discuss our justification in the methods, but based on the reviewer's helpful comment, we have made our justification and reference to this study clearer: 'Specificities at 80% sensitivity, a threshold chosen to be comparable to a prior UK systemic risk factor model which evaluated the performance of a systemic risk factor based prognostic model in individualised DR screening as part of a randomised control trial¹² were determined for comparability.'	Main text: P14 line 354 -356
R3.2	Comment: What is the reason that the multimodal model yield lower prognostic results for the 1-year indicator compared to the pure image model's 1-year prognostic results? The authors are suggested to provide a possible explanation for this discrepancy.	
	Response: In combining the models inputs we take a simple arithmetic mean between the image DLS and tabular DLS. It is possible for the predictions to differ (one is high-risk, and one is not), as the features can differ between images and risk factors.	

	Only the prediction for referable retinopathy (R2+) at the 1-year interval was lower for the multimodal vs image DLS and this was only in the internal test set. It may be that image features for this output / time interval were slightly better at predicting incident referable DR on their own than the mean with the tabular DLS prediction. It is likely that at shorter time predictions, the pre/clinical features are more prominent within images and easier for the image DLS to learn. However, at longer prediction intervals, because image features become more subtle, the multimodal DLS may prevail as the additional risk factors, like duration of DM, provide information not retrievable from images alone.	
R3.3	Comment: The authors have successfully developed and validated DLS for predicting progression for the 1, 2, and 3- year prediction intervals. The prediction of progression within a time period is also a critical problem [1][2], such as progression within 3 years. It would be interesting to know whether the developed DLS model is capable of assessing a patient's risk within a specific time period. [1] Arcadu, F., Benmansour, F., Maunz, A. et al. Deep learning algorithm predicts diabetic retinopathy progression in individual patients. npj Digit. Med. 2, 92 (2019). [2] Bora, Ashish et al. "Predicting the risk of developing diabetic retinopathy using deep learning." The Lancet. Digital health vol. 3,1 (2021): e10-e19. Response: Apologies for any confusion, but if we have understood the comment, then this is in fact what our DLS predict. Bora et al predict if there will be DR onset at any time from the baseline visit to the 2 year interval. We also use the same definition, as we predict if there will be incident referable disease at ANY POINT between the baseline and predict visit hence, 'within' 1 year, 2 years or 3 years. We have now clarified this explicitly in the Methods, where we discuss 'outcomes' definitions: 'The outcome was considered present if it occurred at any point between the baseline and predict visit for the respective interval.'	Main text: P13, line 316-317
R3.4	Comment: The authors used risk factor characteristics alone (tabular DLS), two-field CFPs alone (image DLS) or both for the development of AI model. Does the model incorporate any non- invasive risk factors such as hypertension, triglyceride, low-density lipoprotein cholesterol levels, or renal disease? Response: Our model does not include additional 'non-invasive' risk factors such as blood tests. Biochemistry like LDL/TG/Cholesterol and renal markers are not performed by DR screening services in the UK. Hence they were not available to us, and would likely not be available were the system adopted within the current UK screening pathway. However, we agree this might be an area of future exploration, if it is possible to link our DR screening dataset to hospital/GP data, where these tests could be collected or competed.	
R3.5	Comment: -The EfficientNet-V2 and TabNet models were employed for the analysis of image and tabular data. However, the selection of these models is based on their performance in other related tasks, and the article lacks a sufficient explanation as to why these models are particularly suited for the task at hand, as well as whether other model configurations were attempted.	

	-The methodology lacks a comparison with other baseline methods, which could provide further validation of the proposed approach's effectiveness. Response: Thank you for the helpful comments which we have addressed with changes to the main text. As well as the current references showing the performance of EfficientNet family on other tasks, there are two papers (1,2) which both concluded on the closely related task of DR classification after comparing a large number of other models (VGG, Xception, InceptionV3, InceptionResNetV2, DenseNets, ResNets, NASNets), that EfficientNet models had the best performance on this task.  1. Pak, A., et al. (2020). Comparative analysis of deep learning methods of detection of diabetic retinopathy. Cogent Engineering, 7(1). 2. Das D, Biswas SK, Bandyopadhyay S. Detection of Diabetic Retinopathy using Convolutional Neural Networks for Feature Extraction and Classification (DRFEC). Multimed Tools Appl. 2022 Nov 29:1-59. Early experiments also informed our tabular DLS choice. Namely we trained both a fully connected network and a non-DL based model (Adaboost) to predict incident referable DR/maculopathy within 2 years. Internal test results showed an ensemble of the image DLS with TabNet as the tabular DLS performed the best (AUROC 0.845) compared to a full connected network (AUROC 0.840) or Adaboost (AUROC 0.826). Based on the reviewer's comments, we have added additional justification for the model choice and Tabular DLS ensemble experiments to the supplementary material: Supplementary: 'Additionally, prior studies have demonstrated that EfficientNet models have better performance than a number of other models (VGG, Xception, InceptionV3, InceptionResNetV2, DenseNets, ResNets, NASNets) on the adjacent task of DR classification.10,11 The tabular DLS were based on TabNet, a high-performance neural network for tabular data learning that can reportedly outperforms decision tree algorithms.12 Early experiments also informed our tabular DLS choice. Namely we trained both a fully connected network and a non-DL based model (Adaboost) to predict incident referable DR/maculopathy within 2 years. Internal test results showed an ensemble of the image DLS with TabNet as the tabular DLS performed the best (AUROC 0.845) compared to an ensemble with a full connected network (AUROC 0.840) or Adabaoost (AUROC 0.826).' We also reference this justification in the Supplementary in the methods section of the main text.: 'Justification for the chosen image and tabular DLS models are provided in the supplementary pp 4.'	Supplementary: P4 line 108-110 and 115-120 Main text: P13 line 325-326
R3.6	Comment: The use of two feature extraction networks (EfficientNetV2) that do not share parameters to process multi-view images lacks clear support. This approach	

	could potentially overlook the interrelationships and consistency between different eye images, which is important for disease diagnosis. Response: The decision to use independent networks for each field stems from the clinical observation that referable DR and referable maculopathy affect the macula and nasal field differently (there are exudates more commonly in the macula than in the nasal field in maculopathy for example). Additionally, we hypothesised features may stem from changes in structures in the macula field (fovea) which are not present in the nasal field. Therefore, although there may be shared features in each field (microaneurysms) which could be learned by one shared model, having models specialised to each field is also advantageous. It may be the case that one field has more image features for progression than the other, and this seems supported by our ablation and attribution plots, which show the central macula is important to predictions. Finally, we did also explicitly test this hypothesis during the pre-training (DR severity classification), whereby we trained one EfficientNet-V2s model, but concatenated images at the input of the model (keeping 512px per image) which we call 2-field/1-model vs our current approach of using two models per field which we term 2-field/2-model. We found similar performance in regard to detection of referable DR/maculopathy detection (AUC 0.971 vs 0.972 for 2-field/1-model vs 2-field/2-models). For the reasons detailed above we went for the latter approach, hypothesising learning individual features per field may be advantageous. We have expanded on this justification in the pre-training section of the supplementary and added the above data on 2-field/1-model vs 2-field/2-model AUROC for DR classification as additional data: 'Our early experiments also showed that on the adjacent task of DR detection the use of 2 models independently for each image field was equal to using one model for both imaged fields (concatenated at the input) (referable DR/maculopathy detection AUROC 0.971 vs 0.972 for 2-field/1-model vs 2-field/2-models approaches). However the 2-field, 2-model approach was potentially advantageous because if there are independent prognostic features in each imaged field, but which differed between fields, each model could specialise to each field to learn these features specific to the given field.'	Supplementary: P4 line 110-115
R3.7	Comment: The authors employed RMSProp as the optimization method for pretraining, whereas used AdamW for longitudinal training. Could you provide some insights or references explaining the reason of this setting for the benefit of readers? Response: Thank you for this comment. When we performed the pre-training, we used the same optimiser as described the initial EfficientNet paper (https://arxiv.org/pdf/2104.00298). However, during longitudinal DLS training, we tried both RMSProp and AdamW, and found it was consistently performing better that RMSProp on our tuning set at the 2-year prediction interval. Hence we decided to use this optimiser for all longitudinal training	Supplementary: P5 line 155 and line 192-193

	we did not apply this change retrospectively and redo the pretraining. We have added this to the supplementary DLS pretraining section as follows: 'As the original optimiser used to develop EfficientNet-V2, RMSProp with momentum 0.9 were used ...' 'AdamW was chosen over RMSProp as it consistently outperformed the latter optimiser on early experiments on the 2-year prediction interval.'	
R3.8	Comment: Annotations could be added to subplots a and b in Table 1 to better illustrate the process of constructing the dataset. Response: Thank you for the helpful suggestion. We have added additional detail to the captions of both Table 1 and Figure 1, to better guide readers on how the dataset is constructed, and how Figure 1 relates to longitudinal cohorts in Table 1.	Main text: Figure 1 Captions Table 1: Captions
R3.9	Comment: Line 88: There appears to be a missing closing parenthesis. Response: Thank you for pointing out the error, which has been amended in the main text.	Main text: P4 line 88
R3.10	Comment: Line 121: there is an extraneous 's' present. Response: Thank you for pointing out the error; we have amended this in the main text.	Main text: P5 line 121

REVIEWERS' COMMENTS:

Reviewer #1 (Remarks to the Author):

The authors have answered all the queries nicely. Although the prediction of DR using AI is not a novel concept, the specific application of predicting time-based onset using a multimodal approach that incorporates clinical demographics and fundal images in a single model is less common. Studies such as those conducted by Rom et al. and Dai et al. have investigated comparable territories; however, they lack the same scope or methodological framework.

Reviewer #2 (Remarks to the Author):

The authors responded well to all comments raised but for R2.12 must acknowledge that all their subgroups (age, sex and ethnicity) had small numbers with serious disease (not just for other smaller subgroups, not sure which these would be) and is an important limitation of this study.

Reviewer #3 (Remarks to the Author):

The authors have added information that improves and clarifies the manuscript. No additional suggestions.

Response To Reviewer Comments

Reviewer 2:

#	Comment & Response	Changes in Manuscript
R2.12	Comment: The authors responded well to all comments raised but for R2.12, must acknowledge that all their subgroups (age, sex and ethnicity) had small numbers with serious disease (not just for other smaller subgroups, not sure which these would be) and is an important limitation of this study. Response: We have made additional amendments to the paper to qualify our statements on the subgroup analyses to include all subgroups. We removed line 'and largely in smaller subgroups with few emergent disease cases' in the results when discussing subgroup analyses. The discussion referencing subgroup analyses now reads 'The rarity of emergent disease, in particular referable DR, makes it challenging to draw robust conclusions for the subgroup analyses. Reduced DLS performance in the subgroup analyses should be considered conservative estimates due to the limited positive cases as a result of the low incidence of emergent referable disease.' Finally in the penultimate paragraph on future work, we reference the need for larger validation studies to address limitation with subgroup analyses as follows. 'Larger validation populations would also allow more robust conclusions on the performance of the prognostic DLS, including within demographic subgroups.'	Main text: P10 line 241, P13 line 303-306 and P16 line 371-372